# Evaluation of the Food Choice Motives before and during the COVID-19 Pandemic: A Cross-Sectional Study of 1232 Adults from Croatia

**DOI:** 10.3390/nu13093165

**Published:** 2021-09-10

**Authors:** Tamara Sorić, Ivona Brodić, Elly Mertens, Diana Sagastume, Ivan Dolanc, Antonija Jonjić, Eva Anđela Delale, Mladen Mavar, Saša Missoni, José L. Peñalvo, Miran Čoklo

**Affiliations:** 1Psychiatric Hospital Ugljan, Otočkih Dragovoljaca 42, 23275 Ugljan, Croatia; ravnatelj@pbu.hr; 2Nutrition ID Ltd., Vranovina 30, 10000 Zagreb, Croatia; ivona.nutricionist@gmail.com; 3Unit of Non-Communicable Diseases, Institute of Tropical Medicine, Nationalestraat 155, 2000 Antwerp, Belgium; ellymertens@itg.be (E.M.); dsagastume@itg.be (D.S.); jpenalvo@itg.be (J.L.P.); 4Centre for Applied Bioanthropology, Institute for Anthropological Research, Ljudevita Gaja 32, 10000 Zagreb, Croatia; ivan.dolanc@inantro.hr (I.D.); antonija.jonjic@inantro.hr (A.J.); miran.coklo@inantro.hr (M.Č.); 5Institute for Anthropological Research, Ljudevita Gaja 32, 10000 Zagreb, Croatia; eva.andela.delale@inantro.hr (E.A.D.); sasa.missoni@inantro.hr (S.M.)

**Keywords:** COVID-19, Croatia, determinants, food choice, Food Choice Questionnaire (FCQ), lockdown, motives, SARS-CoV-2

## Abstract

The coronavirus disease 2019 pandemic brought changes to almost every segment of our lives, including dietary habits. We present one among several studies, and the first on the Croatian population, aiming at investigating changes of food choice motives before and during the pandemic. The study was performed in June 2021 as an online-based survey, using a 36-item Food Choice Questionnaire applied for both the periods before and during the pandemic. The final sample consisted of 1232 adults living in Croatia. Sensory appeal was ranked as the number one most important food choice motive before, whereas health was ranked as the number one most important food choice motive during the pandemic. Ethical concern was reported as the least important food choice motive both before and during the pandemic. In women, natural content (*p* = 0.002), health, convenience, price, weight control, familiarity, and ethical concern (all *p* < 0.001) became more important during the pandemic, while price (*p* = 0.009), weight control, familiarity, and ethical concern (all *p* < 0.001) became more relevant for men. All together, these can be considered favorable changes toward optimal diets and may result in beneficial influences on health and lifestyle. Education strategies and efficiently tackling misinformation are prerequisites for informed food choice, which will ensure long-lasting positive effects of such changes.

## 1. Introduction

Nutrition and nutritional habits are recently gaining more attention. Besides the well-known role of unhealthy diet as one of the major risk factors for the development of chronic non-communicable diseases (such as obesity, diabetes, hypertension, metabolic syndrome, etc.) [1], in the context of coronavirus disease 2019 (COVID-19), caused by the SARS-CoV-2 virus in the human population, it has been shown that both forms of malnutrition, i.e., malnourishment (mainly chronic micronutrient deficiency) [2] and obesity [3], can influence the outcomes and the severity of the clinical course in COVID-19 patients. Thus, improving the nutritional status and consuming a healthy balanced diet can be regarded as potential preventive measures against the poor clinical course and fatal outcome of COVID-19.

Since 11 March 2020, when the COVID-19 pandemic was declared by the World Health Organization (WHO) [4], various types and extents of COVID-19 lockdown have occurred all over the world, resulting in drastic and unprecedented changes of all aspects of human life, including dietary habits. Simultaneously, due to the decrease in physical activity levels and an increase in sedentary behavior as consequences of the lockdown [5], weight control became another pressing concern.

Studies have shown different influences of COVID-19 pandemic and lockdown on the diet quality, with some toward a positive influence, while others toward a negative one or no influence. A study conducted in France reported that in some participants, there was an improvement of the diet quality, while in others, the diet quality worsened or did not change [6]. Another study on adults from Quebec, Canada, showed a slight amelioration of the diet quality during the early lockdown [7]. Contrarily, the cross-sectional study conducted by Alhusseini and Alqahtani [8] concluded that the quality of food intake of Saudi Arabian adults deteriorated during the pandemic. The inconsistency of the results on the impact of the COVID-19 pandemic on the diet quality was also confirmed by a scoping review conducted by Bennett et al. [9]. The limitation among these studies lies in the fact that changes in diet quality are just a consequence of an actual change in the food choice motives, and this latter is still not elucidated. This concept is important because it provides a basis for influencing diet quality in a more efficient and long-lasting manner. The relationship between nutritional quality of diet and food choice motives is rather complex, as changes in food choice motives can lead to the increase (e.g., the increase in importance of weight control), decrease (e.g., the increase in importance of mood), or may not have an impact on the nutritional quality of diet (e.g., food choice motives of a person that is not purchasing food for the household).

Studies dealing with food choice motives during the COVID-19 pandemic are still relatively rare. In addition, they tend to analyze other topics such as food planning, shopping, and preparation, diet quality, eating behaviors, the relationship with emotional overeating, body mass index (BMI), and perceived stress [10,11,12,13,14,15], hence food choice motives are addressed only partially and not thoroughly enough.

There are few studies focusing on the difference in food choice motives before and during the COVID-19 pandemic. A study carried out on a population of Polish adolescents [16] concluded that health and weight control were more important, while mood and sensory appeal were found to be less important during the COVID-19 pandemic when compared to the period before it. Another study on French adults showed an increase in weight control, mood, health, sensory appeal, ethical concern, and natural content as food choice motives during the COVID-19 lockdown. Simultaneously, the importance of convenience, familiarity, and price decreased [17]. Similarly, in the study of Snuggs and McGregor [18], the participants placed less importance on familiarity, and more importance on weight control, health, and mood after lockdown when compared to the period before it. Results of these studies suggest significant changes in nutritional habits related to the COVID-19 pandemic and lockdown; however, authors emphasize the necessity of further studies for more robust conclusions.

Generally, habits are automatic behaviors representing a link between the specific context and behavior in response to it [19]. Habits are difficult to change, and, in an evolutionary context, they represent an extremely useful adjustment since they simplify and shorten the response time to repetitive situations, making more complex non-automatic processes of decision making unnecessary. However, when the context changes, non-automatic processes of decision making are engaged [20]. Nutritional habits, including food choices, represent a result of a person-specific “mixture” of various factors, including personal, cultural, socioeconomic, biological (genetic background, gut microbiota through gut-brain axis, etc.), and toxicological (toxicological burden, concentrations of essential micronutrients, etc.) factors. Integration of such factors and their levels of priority results in food choice motives, which are showing a large inter-individual variability and are playing a major role in shaping food choice decisions [21]. In a significantly changed environment, such as the COVID-19 pandemic and lockdown, we hypothesized that such “mixture” is being “re-mixed” to suit the new set of priorities. Consequently, such a change would result in changes in food choice motives and finally in those of the overall diet quality. Therefore, the aim of the present study was to investigate the changes in food choice motives before and during the COVID-19 pandemic in a study population of adults from Croatia. Since there are consistent sex differences in the area of food choice [22], special emphasis in the present study was put on the difference in food choice motives between men and women.

## 2. Materials and Methods

### 2.1. Study Design and Study Population

This cross-sectional study is a part of the international project aiming at investigating the impact of the COVID-19 pandemic on food choice motives in the adult population of Croatia and Belgium. In this paper, we presented the results of the Croatian part of the research.

The present study was conducted using an anonymous online questionnaire designed in Google Forms online survey platform. The data collection was carried out after the third wave of COVID-19 in Croatia, i.e., from 16 June 2021 to 30 June 2021. At that time, more or less the same counter-epidemic measures were imposed as before, without strict lockdown. During this period, research team members distributed invitations to participate in the research to their professional networks and personal contacts by e-mail, messaging applications (WhatsApp^®^, Viber^®^, and Messenger^®^), social media networks (Facebook^®^, Instagram^®^, and LinkedIn^®^), and by posting research advertisements in relevant groups on social media networks (e.g., public Facebook^®^ groups aimed at gathering students, those formed for the purposes of survey exchange, etc.). To increase the number of study participants, the research team members asked the potential participants to further forward the invitations to their contacts. The reason for choosing a non-probabilistic sample was mainly influenced by a greater possibility of gathering a relatively large sample during a short period of time, which was important due to the unpredictable nature of the COVID-19 pandemic.

Participants aged 18 years and older and residing in Croatia were considered eligible to enter the study. A total of 1250 participants completed the online questionnaire. Following the end of the data collection period, study data were checked to verify if there were any missing data and/or duplicate, inappropriate, or inaccurate answers. The data verification process resulted in the exclusion of 18 participants (1.4%) for the following set of reasons: being under the age of 18 (*n* = 3), multiple submission (*n* = 1), a personal request made by the participant who has completed the questionnaire but has subsequently notified the research team member about not currently residing in Croatia (*n* = 1), reported implausible age, body weight, and/or body height (*n* = 13). Therefore, the final sample consisted of 1232 participants.

### 2.2. Ethical Issues

All the participants were included in the present study voluntarily. In the participant information sheet, at the beginning of the online questionnaire, the potential participants were provided with detailed information on the main outcomes of the research, estimated time needed for the completion of the questionnaire (~15 min), and the potential benefits that might result from the participation in the research (completing the questionnaire could have given the participants a more detailed insight in the food choice motives that dominate their food procurement and it could have helped them become more aware of their dietary habits that could ultimately motivate them to make positive changes in their diet). By clicking on the “Next” button at the end of the participant information sheet, the participants confirmed to have read the participant information sheet and agreed to participate in the present scientific research. Only after providing the informed consent were participants able to proceed with the completion of the questionnaire. All the participants had the right to withdraw from the research at any time without consequences (the responses of each participant were saved only after clicking on the “Submit” button at the end of the online questionnaire). There were no harmful health effects that could be caused by participation in the present study. The participants have not received financial or any other compensation for their participation in the research.

### 2.3. Measures and Outcomes

The research data were collected using a structured online questionnaire that consisted of a total of 87 questions divided into three sections.

#### 2.3.1. Socio-Demographic, COVID-19, and Anthropometric Data

The first section consisted of 15 questions designed to gather all the relevant socio-demographic information (seven mandatory and one non-mandatory question), information on COVID-19 (four mandatory and one non-mandatory question), and relevant anthropometric measures (two mandatory questions). Self-reported body weight and height were used to calculate BMI, i.e., weight in kilograms divided by height in meters squared. Participants’ characteristics from the first section of the applied questionnaire are listed in Table 1.

#### 2.3.2. Food Choice Questionnaire

Changes in motives influencing food choice before and during the COVID-19 pandemic were evaluated using the Western Balkan Countries (WBCs) version of the Food Choice Questionnaire (FCQ) [23], originally developed by Steptoe et al. [24]. Since then, the FCQ has been widely used in numerous scientific research studies conducted in different population groups, and several modified versions of the original FCQ, as the one used in the present study, have been developed thus far [23,25,26,27,28]. Prior to the study start, the research team received written permissions for the use of the FCQ from both the authors of the original FCQ and the authors of the WBCs version of the FCQ. For the purposes of the present study, some of the items from the WBCs version of the FCQ were slightly modified with more appropriate translations to the Croatian language.

The original 36-items FCQ was designed to assess health and non-health motives influencing food choices, grouped into nine factors, namely: Health (six items related to chronic disease prevention, general nutrition, and well-being); Mood (six items related to stress control, relaxation, general alertness, and mood); Convenience (five items concerning the purchase and preparation of food); Sensory Appeal (four items associated with smell, taste, texture, and appearance of food); Natural Content (three items reflecting concern with the content of additives and artificial ingredients); Price (three items associated with the cost of food); Weight Control (three items related to consumption of food low in calories and fat); Familiarity (three items asking how important it is for the participants to eat the food they are used to); and Ethical Concern (three items related to environmental and political issues) [24].

In the present study, the participants were asked to rate the extent to which they agree or disagree with the 36 FCQ items twice: once to assess their general standings toward food choices before the COVID-19 pandemic (the second section of the questionnaire) and once to assess their standings toward food choices during the COVID-19 pandemic (the third section of the questionnaire). Each of the statements started with “It is important to me that the food I eat on a typical day …” and the importance of each item was estimated based on the choice of one response on a five-point Likert scale. The available answers were as follows: 1—strongly disagree; 2—disagree; 3—neither agree nor disagree; 4—agree; and 5—strongly agree. Even though the original FCQ had a four-point scale [24], the WBCs version of the FCQ used a five-point scale, particularly for the purpose of preventing artificial agreements and/or disagreements of the study participants [23].

### 2.4. Statistical Analysis

General characteristics of the study participants (socio-demographic characteristics, information on COVID-19, and relevant anthropometric measures) were presented as means and standard deviations for continuous variables and as numbers of participants and percentages for categorical variables. For testing sex differences (men vs. women) within categorical variables Pearson chi-square test, with Yates’ corrections for small values in cells, was used. A *t*-test for independent samples was used for testing sex differences within continuous variables.

Based on the a priori theory of food choice motives, previous research [23,24], and the changes compared to both the original FCQ and the WBCs version of the FCQ, we turned to exploratory factor analysis in order to define the loadings for the items included within the factors. It was conducted using the maximum likelihood estimation method and varimax rotation. Item and factor descriptive-statistical indicators (means and standard deviations) were calculated to establish the psychometric properties of the FCQ for the total sample, as well as for sex stratification, for both the periods before and during the COVID-19 pandemic. To verify the internal reliability of the data within assessed factors, Cronbach’s alpha coefficients were calculated for the nine FCQ factors for both measurement times (before and during the COVID-19 pandemic).

Prior to data analysis, the data were examined for normal distributions. Despite significantly negatively skewed distributions of results on factors and most of the items, confirmed by the Shapiro–Wilk test (all at *p* < 0.01 at both measurement times), since this is a large sample, variables vary in the same directions and based on the value of the skewness and kurtosis statistics [29,30] the sex differences were compared by using a one-way analysis of variance (a one-way ANOVA). An ANOVA with repeated measures was used to compare the results obtained for the periods before and during the COVID-19 pandemic.

The Statistical Package for the Social Sciences (SPSS), version 13.0. was used for the data analysis. The level of significance was set at *p* < 0.05.

## 3. Results

The present study was conducted on a non-probabilistic sample of Croatian adults, with the vast majority of the study participants being female (72.2%). The age of the study participants ranged from 18 to 87 years (median = 33). Overall, participants’ general characteristics and stratified by sex are summarized in Table 1. Regarding the socio-demographic characteristics, men and women significantly differed in terms of employment status, with more female than male students (*p* < 0.001). Furthermore, significantly more employed women, when compared to employed men, worked or were closely related to the healthcare system (*p* < 0.001). Moreover, men reported having significantly higher average monthly net incomes when compared to women (*p* < 0.001). Significantly more men, when compared to women, have been infected with COVID-19 (*p* = 0.005). As for the BMI, 34.3% of the total sample was overweight or obese (24.9% women vs. 58.7% men, *p* < 0.001).

Table 2 summarizes the factor analysis loadings performed on the 36-item FCQ. Instead of the original nine, exploratory factor analyses extracted seven factors before the COVID-19 pandemic and six factors during the COVID-19 pandemic (eigenvalues higher than 1). This structure did not confirm the original structure of factors from Steptoe et al. [24] or the later structure within the WBCs study [23]. After various solutions, the approach based on theoretical background and many years of use of the FCQ was found to be the most appropriate to interpret. Varimax rotation with a fixed number of factors (nine) was performed and mostly confirmed the original structure of factors from Steptoe et al. [24], and it was a better fit than the WBCs structure [23]. The nine factors together accounted for 65.2% of the variance before and 71.0% during the COVID-19 pandemic. In both factor structures, the differences from the normative sample were similar to those observed in other studies, e.g., health shared variance with natural content as it was in Belgium and Italy [31] and WBCs [23]. The items that cross-loaded on other factors (>0.50 on other factors) are also indicated in Table 2. The only significant difference from previous studies was the item “Is like the food I ate when I was a child”, loaded onto the ethical concern factor, within both before (0.462) and during the COVID-19 pandemic (0.431) factor structure, which was one of the reasons to interpret factors according to the original structure [24] and their content validity. The results of the loadings and Cronbach’s alpha coefficients for the applied FCQ for both the periods before and during the COVID-19 pandemic for the studied sample (*n* = 1232) are listed in Table 2. The reliability for most of the nine factors was excellent or good (Cronbach’s alpha higher than 0.80); only familiarity possessed a Cronbach’s alpha of 0.69.

The importance of motives influencing food choice before and during the COVID-19 pandemic for the overall studied sample and sex stratification are shown in Table 3. The higher mean indicates the higher importance of each motive. At the level of the overall study sample, the participants reported that before the COVID-19 pandemic, it was the most important for the food they eat on a typical day to keep them healthy, to taste good, to contain natural ingredients, and to represent a good value for money (all means ≥ 4.00). For both men and women, the most important food choice motive both before and during the COVID-19 pandemic was that it tastes good (the highest mean score), while the least important was that it comes from the countries they approve of politically (the lowest mean score). For both sex, the importance of numerous food choice motives significantly increased during the COVID-19 pandemic when compared to the period before it (Table 3). Before the COVID-19 pandemic, all the food choice motives, except for the fact that the food they eat on a typical day is low in calories and fat, is familiar, cheers them up, helps them control their weight, helps them relax, is high in protein, is what they usually eat, and helps them cope with life (all *p* > 0.05), were significantly more important to women when compared to men. Similarly, during the COVID-19 pandemic, all the food choice motives were significantly more important to women when compared to men, with the exception of the food they eat on a typical day being low in calories and fat, helping them control their weight, being high in protein, being what they usually eat, and helping them cope with life (all *p* > 0.05) (Table 3).

Table 4 shows the importance of factors influencing food choice before and during the COVID-19 pandemic for the total sample and sex stratification. At the level of the overall study sample, the most important factors influencing food choice both before and during the COVID-19 pandemic were health, sensory appeal, and convenience (with sensory appeal ranked as the number one most important before the pandemic and health ranked as the number one most important during the pandemic), while ethical concern remained the least important one. In women, natural content (*p* = 0.002), health, convenience, price, weight control, familiarity, and ethical concern (all *p* < 0.001) were reported to be more important during the COVID-19 pandemic when compared to the period before the pandemic. Men declared price (*p* = 0.009), weight control, familiarity, and ethical concern (all *p* < 0.001) to be more important during the COVID-19 pandemic when compared to the period before it. When compared to men, the vast majority of the factors was more important to women both before and during the COVID-19 pandemic, with the exception of weight control for which there was no statistically significant difference neither before nor during the pandemic and familiarity that was significantly more important to women only during the pandemic.

## 4. Discussion

The present cross-sectional study identified health, sensory appeal, and convenience as the most important factors influencing food choice before and during the COVID-19 pandemic, specifically with sensory appeal ranked as the number one most important before the pandemic and health ranked as the number one most important during the pandemic. Both before and during the COVID-19 pandemic, ethical concern was revealed to be the least important factor when making food choices. At the subgroup level, the importance of numerous factors influencing food choice increased during the pandemic in both men and women, with health, mood, convenience, sensory appeal, natural content, price, familiarity, and ethical concern being more important to women. To date and to the best of our knowledge, there were only a few previously published studies comparing the importance of food choice motives before and during these challenging times using the original FCQ [16,18] or its slightly modified version [17], with the study at hand being the first one evaluating this among Croatian adults.

Since the very beginning of the COVID-19 pandemic, the importance of proper nutrition, as a requirement for a healthy immune system, has been widely emphasized [32,33,34,35]. Therefore, it was not surprising for the results to have shown that the importance of health, natural content, and weight control significantly increased among women during the COVID-19 pandemic when compared to the period before it. Among those three factors, only the importance of weight control increased during the pandemic in men. The increase in the importance of weight control in both men and women could potentially be explained by the fact that COVID-19 lockdown, and the pandemic in general, were closely associated with the increase in sedentary lifestyle and weight gain, which was confirmed by numerous previously published studies conducted on different population groups [36,37,38,39,40,41]. For the interpretation of the aforementioned results of the study at hand, it is worthy of mentioning a study by Đogaš et al. [42] that confirmed that a total of 30.7% of the study participants gained weight during the spring 2020 COVID-19 lockdown in Croatia, with women reporting lower frequency and duration of physical activity, which was potentially one of the main reasons leading to their weight gain. Nowadays, during the later phase of the pandemic, people are probably more aware of their weight gain and have, therefore, started to pay more attention to weight control.

On the other hand, the importance of mood and sensory appeal on food choice has not changed significantly during the COVID-19 pandemic when compared to the period before it, neither in men nor in women. In a more detailed analysis, based on the items included within the mood factor, for both men and women, it became more important for the food they eat on a typical day to help them cope with stress. The COVID-19 confinement consequently led to the increase in stress, fear, anxiety, and numerous other mental health issues [43,44,45,46,47], which was also confirmed by some previous Croatian studies showing higher frequency of fear, discouragement, and sadness [42] as well as increased stress perception [48]. It is likely that this has prompted people to look for different ways that could help them cope with those problems, one of which certainly is food choice.

When focusing on convenience, the importance of this single factor increased during the pandemic among women, and the results have also confirmed higher importance among women when compared to men, both before and during the pandemic. These findings are aligned with the results of the studies of Daniels et al. [49] and Achón et al. [50], who have confirmed that women spend more time cooking and grocery shopping when compared to men.

As already mentioned, the vast majority of the studied factors turned out to be significantly more important to women when compared to men. Such findings are being supported by the results of the population-based study conducted on a representative sample of Finnish adults revealing that women, when compared to men, pay more attention to potential pandemics and are more focused on how the food affects their health [51].

Similarly to the results of the study at hand, in the study conducted on a sample of 2448 Polish adolescents (1552 females and 896 males), among females, the importance of health, natural content, and weight control increased during the COVID-19 pandemic, when compared to the period before it [16]. In male participants of both the present study and the aforementioned Polish adolescents’ study [16], the importance of a smaller number of factors changed significantly when compared to females. Likewise, in another study that examined the changes in food choice motives before and during the COVID-19 lockdown in France, the importance of weight control, mood, health, sensory appeal, ethical concern, and natural content increased, while the importance of familiarity, price, and convenience decreased [17]. For the interpretation and comparison with the results of the present study, it is important to highlight the fact that in the French study, a French version of the FCQ, including a total of 24 items, was used. Additionally, before the COVID-19 pandemic, sensory appeal was ranked as the number one most important, and the ethical concern as the least important food choice motive in the studied population. This is in line with the results of a systematic review conducted by Cunha et al. [52], including, among others, individuals from Croatia.

Prioritizing the food choice motives associated with health and weight control is beneficial not only on the personal level but also for society in general. Therefore, the results of this and similar studies are useful and relevant not only for the scientific community but for all the involved stakeholders, including policymakers, food producers, and educators. Changes in food choice motives during the COVID-19 pandemic, as a long-term crisis going on for more than a year now, inevitably influence all aspects of life, including cultural, socioeconomic, ecological, political, scientific, medical, etc. Among others, as already mentioned by Głąbska et al. [16], the COVID-19 pandemic influences social interactions in a way that even social stigmatization and prejudice can easily occur, resulting in the marginalization of individuals and sub-populations based on their priorities in food choice motives. Therefore, the results of this and similar studies will allow the creation of more appropriate and effective dietary policies based on their adjustment at the (sub-)population level and taking into consideration their cultural background and specificities.

As for the socioeconomic influences of the COVID-19 pandemic and associated changes in food choice motives, health becoming the number one most important motive during the pandemic is expected to lead to an increase in the market share of organic and local products. An increase in demand for foods having health-promoting properties, being low in calories and fat, and being perceived as more valuable could also be expected. All of these factors can influence the change of general structure of consumption, positions on national and international markets, increase in the market share of healthy food products, and decrease in those products that are perceived as not being health beneficial [16].

COVID-19 crisis can also be seen as an opportunity to improve eating behaviors and create desirable and healthy eating habits. Public health policymakers should therefore pursue public health actions that take into consideration knowledge on which food choice motive changes can increase or decrease the nutritional quality of diet. Although changes in food choice motives might result in a change of nutritional behaviors of people and their nutritional quality, it might also represent a more efficient way of prevention of diet-related chronic non-communicable diseases. Therefore, it is also important to explore changes in health status at both individual and population levels as a possible final consequence of changes in food choice motives.

When looking at the broader societal context and implications of changes of food choice motives, it is very important to also consider the possibility of such changes being only temporary and restored to a pre-lockdown state once the crisis is over. Successful educational programs and strategies are therefore crucial in trying to make such favorable changes long-lasting. When speaking more specifically about Croatia, according to the Croatian Institute of Public Health, almost two-thirds of the adult population are overweight or obese [53]. For that reason, various educational programs and public health intervention strategies at the local and/or regional levels were and are being conducted. The most important such program at the national level is an EU-funded program, “Living Healthy”, coordinated by the Croatian Institute of Public Health [54]. The program itself includes both educational components and dietary interventions. In addition, adequately fighting misinformation (that could, in terms of food quality and food choice motives, include attributing positive or negative properties of certain foods in regard to their impact on primary or secondary prevention and/or treatment of COVID-19) and providing complete, reliable, and timely information are necessary prerequisites for informed food choice.

### 4.1. Study Limitations and Strengths

In the present study, the participants were asked to retroactively evaluate the importance of food choice motives for the period before the COVID-19 pandemic, which began more than a year before this study was conducted. In the context of the results for the aforementioned period, a possibility of recall bias, which includes a greater possibility of providing insufficiently accurate estimates, cannot be excluded and represents the most important study limitation. Such a recall bias was inevitable since the pandemic was unpredictable in its beginning and length, and it still is in its course, so it was impossible to organize a prospective data collection before the pandemic. The fact that the type of the sample we used was non-probabilistic, resulting, for example, in the vast majority of the study participants being female, inevitably and significantly compromises its representativeness. Another limitation is related to the extent of possible extrapolation of the results since, however detailed it is, it is still a study on one population (Croatian). In addition, it is a study on an adult population, which makes it hard to extrapolate results to, for example, adolescent populations. As this study used an online-based survey with anonymous participants, it makes it impossible to exclude to a certain extent a possible information reliability bias. Although we conducted the data verification process, due to restrictions imposed because of the counter-epidemic measures, it was actually impossible to avoid this bias. On the personal level of each participant, we cannot exclude possible events and/or changes in his/her life that are not directly related to the COVID-19 pandemic (e.g., pregnancy) but could have influenced food choice motive changes. Along with the occurrences not directly related to the current pandemic, it is also important to highlight the fact that the pandemic has affected individuals differently (e.g., in terms of workload and the possibility of working remotely). Since we cannot be aware of such situations, it is a study limitation that could not be prevented and/or mitigated.

The facts that, to the best of our knowledge, there are no similar studies so far for the adult Croatian population, and overall, only a few studies compare food choice motives before and during the COVID-19 pandemic using the FCQ as a tool previously proven to be reliable, represent major study strengths and its scientific novelty.

### 4.2. Perspectives for the Future Studies

The main idea of the whole international project CFC CRO-BE, to compare the results of the changes in food choice motives before and during the COVID-19 pandemic from Croatia and Belgium, is based on their differences (especially in geographical position, climate, people mentality, and their habits), but at the same time on some interesting similarities. The aforementioned would result in internationally relevant data and might represent a basis for the appropriate policy changes for facing this and future similar crises. The study conducted in Belgium, as a part of the CFC CRO-BE project, could be seen as a pilot testing of this international context, which could further be expanded to other countries and (sub-)populations as well.

Future follow-up studies should evaluate whether changes to food choice motives during the COVID-19 pandemic are long-lasting or tend to return to the pre-pandemic state. This would represent a quality basis for adequate changes in educational strategies and strategies for tackling misinformation and the promotion of informed food choices. It is also important to investigate whether the change to the nutritional quality of the diet resulted in health effects, such as reduction in the prevalence of complex but diet-related diseases (obesity, diabetes, hypertension, metabolic syndrome, etc.) and/or changes in lifestyle (increase in physical activity, favorable changes of nutritional habits, etc). This would represent a basis for the development of more efficient public health policies related to the prevention of diet-related diseases.

## 5. Conclusions

To the best of our knowledge, the present study is the first to evaluate the changes in food choice motives before and during the COVID-19 pandemic among Croatian adults. Health, sensory appeal, and convenience were considered as the most important food choice motives both before and during the pandemic, with health ranked as the number one most important motive during these challenging times. The importance of numerous factors influencing food choice, including health and natural content in women and weight control in both men and women, increased during the pandemic when compared to the period before it. The majority of the factors were more important to women both before and during the COVID-19 pandemic when compared to men. All together, these changes can be considered favorable and can result in beneficial impacts on health and lifestyle. Education strategies, as well as efficient misinformation tackling, are prerequisites for informed food choice, which will ensure long-lasting positive effects of such changes in food choice motives.

If there can be anything positive about this pandemic, which is being undoubtedly horrible in so many instances, it would be the personal realization that we have to take responsibility for what we eat, how we live, and for our health.

## Figures and Tables

**Table 1 nutrients-13-03165-t001:** General characteristics of the study participants.

Characteristics	Total (*n* = 1232)	Women (*n* = 890)	Men (*n* = 339)	*p*
**Socio-demographic characteristics**				
Age (years), mean (SD)	35.4 (10.35)	35.2 (10.43)	35.8 (10.15)	0.377 ^e^
Educational level ^a^, *n* (%)				
Secondary/high school or less	253 (20.5)	177 (19.8)	76 (22.4)	0.804 ^f^
Bachelor’s degree	197 (16.0)	143 (16.1)	54 (15.9)	
Master’s degree	598 (48.5)	436 (49.0)	160 (47.2)	
Doctorate degree	184 (14.9)	134 (15.1)	49 (14.5)	
Employment status, *n* (%)				
Employed	1019 (82.7)	723 (81.2)	294 (86.7)	<0.001 ^f^
Unemployed	69 (5.6)	52 (5.8)	17 (5.0)	
Retired	19 (1.5)	13 (1.5)	6 (1.8)	
Student	125 (10.1)	102 (11.5)	22 (6.5)	
Working in or being closely related to the healthcare system ^b^, *n* (%)	268 (26.4)	212 (29.4)	55 (18.8)	<0.001 ^f^
Marital status, *n* (%)				
Unmarried	605 (49.1)	438 (49.2)	165 (48.7)	0.492 ^g^
Married	567 (46.0)	404 (45.4)	162 (47.8)	
Divorced	52 (4.2)	40 (4.5)	12 (3.5)	
Widowed	8 (0.6)	8 (0.9)	0 (0.0)	
Residential area, *n* (%)				
Urban area (city)	1091 (88.6)	782 (87.9)	306 (90.3)	0.280 ^g^
Rural area (countryside)	141 (11.4)	108 (12.1)	33 (9.7)	
Average monthly net income ^c^, *n* (%)				
No independent income	98 (8.0)	84 (9.4)	14 (4.1)	<0.001 ^f^
Lower than the minimum net salary	65 (5.3)	54 (6.1)	10 (2.9)	
Between minimum and average net salary	369 (30.0)	302 (33.9)	66 (19.5)	
Higher than the average net salary	700 (56.8)	450 (50.6)	249 (73.5)	
**Information on COVID-19**				
COVID-19 infection, *n* (%)				
Yes	342 (27.8)	228 (25.6)	114 (33.6)	0.005 ^f^
No	890 (72.2)	662 (74.4)	225 (66.4)	
Method used for virus infection confirmation in those who tested positive ^d^, *n* (%)				
PCR test	195 (57.4)	126 (55.8)	69 (60.5)	0.761 ^f^
Rapid antigen test	52 (15.3)	35 (15.5)	17 (14.9)	
Serology test	29 (8.5)	19 (8.4)	10 (8.8)	
Not confirmed by any of the above-mentioned methods	64 (18.8)	46 (20.4)	18 (15.8)	
COVID-19 vaccination, *n* (%)				
Yes	632 (51.3)	445 (50.0)	187 (55.2)	0.106 ^f^
No	600 (48.7)	445 (50.0)	152 (44.8)	
Self-isolation due to COVID-19 preventive measures, *n* (%)				
Yes	484 (39.3)	339 (38.1)	144 (42.5)	0.159 ^f^
No	748 (60.7)	551 (61.9)	195 (57.5)	
Confirmed COVID-19 infection in members of the household, *n* (%)				
Yes	453 (36.8)	313 (35.2)	138 (40.7)	0.071 ^f^
No	779 (63.2)	577 (64.8)	201 (59.3)	
**Anthropometric data**				
BW (kg), mean (SD)	72.2 (16.15)	66.2 (12.60)	87.9 (13.70)	<0.001 ^e^
BH (cm), mean (SD)	172.9 (8.92)	169.0 (6.13)	183.0 (6.98)	<0.001 ^e^
BMI (kg/m^2^), *n* (%)				
Underweight (BMI < 18.5 kg/m^2^)	41 (3.3)	39 (4.4)	2 (0.6)	<0.001 ^f^
Normal BW (18.5–24.9 kg/m^2^)	769 (62.4)	630 (70.8)	138 (40.7)	
Overweight (25.0–29.9 kg/m^2^)	315 (25.6)	158 (17.8)	155 (45.7)	
Obesity (≥30.0 kg/m^2^)	107 (8.7)	63 (7.1)	44 (13.0)	

*n*, number of participants; SD, standard deviation; PCR test, polymerase chain reaction test; BW, body weight; BH, body height; BMI, body mass index; ^a^ the applied questionnaire originally contained six available answers in the question regarding educational level, however for the purposes of statistical analysis, secondary/high school and lower levels of education (a few grades of primary school and finished primary school) were grouped together; ^b^ a non-mandatory question; the calculation was based on the number of participants who have reported to be employed (data for four participants are missing: three women and one man); ^c^ minimum net salary = 3400 Croatian kunas, average net salary = 7000 Croatian kunas; ^d^ a non-mandatory question; the calculation was based on the number of participants who have reported to have had COVID-19 infection (data for two women are missing); ^e^
*t*-test for independent samples; ^f^ Pearson chi-square test; ^g^ Yates’ correction. Three participants (0.2%) have selected the answer “I prefer not to say” in the sex question of the applied questionnaire, which is the reason for the sum of men and women not being equal to the overall sample size.

**Table 2 nutrients-13-03165-t002:** The results of the loadings and Cronbach’s alpha coefficients according to the original nine factors of the Food Choice Questionnaire for the period before and during the COVID-19 pandemic for the studied sample (*n* = 1232).

Items ^a^	Before COVID-19 Pandemic			During COVID-19 Pandemic		
	Loading	Loading besides Original (>0.50 on Other Factors)	α	Loading	Loading besides Original (>0.50 on Other Factors)	α
**Health**						
29. Keeps me healthy	0.806		0.92	0.781		0.93
30. Is good for my skin/teeth/hair/nail, etc.	0.756			0.742		
22. Contains a lot of vitamins and minerals	0.749			0.781		
10. Is nutritious	0.654			0.665		
9. Is high in fiber and roughage	0.634			0.684		
27. Is high in protein	0.606			0.627		
**Mood**						
26. Helps me relax	0.711		0.90	0.809		0.93
34. Helps me to cope with life	0.704			0.738		
16. Helps me cope with stress	0.675			0.800		
31. Makes me feel good	0.540	0.537 Health		0.592		
24. Keeps me awake/alert	0.449			0.590		
13. Cheers me up	0.447	0.559 S. Appeal		0.732		
**Convenience**						
28. Takes no time to prepare	0.781		0.89	0.813		0.92
15. Can be cooked very simply	0.747			0.779		
1. Is easy to prepare	0.697			0.761		
35. Can be bought in shops close to where I live or work	0.486			0.563		
11. Is easily available in shops and supermarkets	0.466			0.562		
**Sensory Appeal**						
14. Smells nice	0.700		0.83	0.718		0.88
4. Tastes good	0.511			0.544		
18. Has a pleasant texture	0.465			0.531		
25. Looks nice	0.405			0.557		
**Ethical Concern**						
20. Comes from countries I approve of politically	0.609		0.75	0.626		0.73
19. Is packaged in environmentally friendly way	0.597			0.560		
32. Has a country of origin clearly marked	0.541			0.557		
**Price**						
36. Is cheap	0.813		0.79	0.764		0.82
6. Is not expensive	0.736			0.747		
12. Is good value for money	0.317			0.358		
**Weight Control**						
3. Is low in calories	0.619		0.79	0.684		0.85
7. Is low in fat	0.581			0.664		
17. Helps me control my weight	0.484			0.547		
**Familiarity**						
33. Is what I usually eat	0.622		0.69	0.694		0.78
8. Is familiar	0.611			0.636		
21. Is like the food I ate when I was a child	0.180			0.271		
**Natural Content**						
2. Contains no additives	0.431		0.86	0.442	0.568 Health	0.89
5. Contains natural ingredients	0.377	0.670 Health		0.301	0.700 Health	
23. Contains no artificial ingredients	0.298	0.566 Health		0.223	0.661 Health	

α, Cronbach’s alpha coefficient, ^a^ the numbers in front of the items represent the ordinal numbers of the items in both the original version of the Food Choice Questionnaire developed by Steptoe et al. [24] and the Western Balkan Countries version of the Food Choice Questionnaire developed by Milošević et al. [23].

**Table 3 nutrients-13-03165-t003:** The motives influencing food choice before and during the COVID-19 pandemic for the total sample and separately with respect to sex.

Items ^a^	Total Sample (*n* = 1232)		Women (*n* = 890)			Men (*n* = 339)			Women vs. Men before COVID-19 Pandemic	Women vs. Men during COVID-19 Pandemic
	Before COVID-19 Pandemic Mean (SD)	During COVID-19 Pandemic Mean (SD)	Before COVID-19 Pandemic Mean (SD)	During COVID-19 Pandemic Mean (SD)	*p* ^b^	Before COVID-19 Pandemic Mean (SD)	During COVID-19 Pandemic Mean (SD)	*p* ^b^	*p* ^c^	*p* ^c^
**Factor 1—Health**										
22. Contains a lot of vitamins and minerals	3.87 (1.015)	3.92 (1.004)	3.95 (0.995)	4.01 (0.954)	0.010	3.65 (1.039)	3.69 (1.091)	0.334	<0.001	<0.001
29. Keeps me healthy	4.11 (0.965)	4.10 (0.967)	4.19 (0.936)	4.19 (0.931)	0.777	3.91 (1.015)	3.87 (1.023)	0.355	<0.001	<0.001
10. Is nutritious	3.99 (0.981)	4.08 (0.934)	4.04 (0.951)	4.16 (0.902)	<0.001	3.84 (1.044)	3.86 (0.982)	0.534	0.001	<0.001
27. Is high in protein	3.65 (1.022)	3.74 (1.028)	3.64 (1.021)	3.74 (1.019)	<0.001	3.67 (1.028)	3.73 (1.058)	0.067	0.655	0.969
30. Is good for my shin/teeth/hair/nail, etc.	3.87 (1.065)	3.93 (1.049)	3.97 (1.038)	4.05 (1.001)	<0.001	3.61 (1.092)	3.61 (1.102)	0.929	<0.001	<0.001
9. Is high in fiber and roughage	3.56 (1.024)	3.71 (1.011)	3.62 (0.987)	3.80 (0.987)	<0.001	3.40 (1.103)	3.49 (1.039)	0.032	0.001	<0.001
**Factor 2—Mood**										
16. Helps me cope with stress	3.11 (1.228)	3.22 (1.210)	3.19 (1.219)	3.27 (1.204)	0.005	2.92 (1.230)	3.09 (1.220)	<0.001	0.001	0.022
34. Helps me to cope with life	3.00 (1.216)	3.10 (1.225)	3.00 (1.212)	3.14 (1.234)	<0.001	2.97 (1.227)	3.00 (1.200)	0.404	0.634	0.081
26. Helps me relax	3.17 (1.125)	3.25 (1.176)	3.19 (1.124)	3.29 (1.171)	<0.001	3.11 (1.125)	3.12 (1.182)	0.947	0.293	0.018
24. Keeps me awake/alert	3.37 (1.123)	3.38 (1.150)	3.43 (1.123)	3.45 (1.149)	0.447	3.20 (1.110)	3.17 (1.129)	0.530	0.001	<0.001
13. Cheers me up	3.72 (1.108)	3.47 (1.139)	3.75 (1.101)	3.52 (1.128)	<0.001	3.66 (1.126)	3.35 (1.160)	<0.001	0.212	0.016
31. Makes me feel good	3.83 (1.060)	3.74 (1.098)	3.89 (1.043)	3.81 (1.082)	0.002	3.66 (1.091)	3.56 (1.122)	0.018	0.001	<0.001
**Factor 3—Convenience**										
1. Is easy to prepare	3.93 (1.027)	3.97 (1.010)	4.04 (0.972)	4.07 (0.977)	0.240	3.64 (1.112)	3.71 (1.050)	0.196	<0.001	<0.001
15. Can be cooked very simply	3.88 (1.002)	3.95 (0.989)	3.99 (0.976)	4.07 (0.945)	<0.001	3.60 (1.017)	3.65 (1.041)	0.174	<0.001	<0.001
28. Takes no time to prepare	3.88 (1.003)	3.90 (1.017)	3.97 (0.975)	4.02 (0.987)	0.060	3.64 (1.037)	3.59 (1.032)	0.207	<0.001	<0.001
35. Can be bought in shops close to where I live or work	3.69 (1.103)	3.77 (1.106)	3.77 (1.086)	3.87 (1.091)	<0.001	3.48 (1.123)	3.48 (1.100)	0.940	<0.001	<0.001
11. Is easily available in shops and supermarkets	3.83 (1.043)	3.93 (1.004)	3.89 (1.017)	4.04 (0.958)	<0.001	3.64 (1.090)	3.62 (1.060)	0.632	<0.001	<0.001
**Factor 4—Sensory Appeal**										
14. Smells nice	3.85 (1.073)	3.87 (1.031)	3.93 (1.054)	3.97 (1.008)	0.173	3.65 (1.098)	3.63 (1.053)	0.724	<0.001	<0.001
25. Looks nice	3.41 (1.118)	3.59 (1.080)	3.47 (1.127)	3.66 (1.089)	<0.001	3.24 (1.078)	3.40 (1.034)	<0.001	0.002	<0.001
18. Has a pleasant texture	3.69 (1.059)	3.71 (1.049)	3.76 (1.047)	3.80 (1.034)	0.130	3.49 (1.067)	3.47 (1.050)	0.568	<0.001	<0.001
4. Tastes good	4.46 (0.890)	4.22 (0.917)	4.52 (0.846)	4.30 (0.881)	<0.001	4.30 (0.983)	4.00 (0.976)	<0.001	<0.001	<0.001
**Factor 5—Natural Content**										
2. Contains no additives	3.53 (1.101)	3.67 (1.090)	3.60 (1.070)	3.76 (1.053)	<0.001	3.35 (1.155)	3.45 (1.148)	0.035	<0.001	<0.001
5. Contains natural ingredients	4.08 (0.947)	4.01 (0.961)	4.12 (0.928)	4.07 (0.920)	0.034	3.97 (0.977)	3.84 (1.034)	0.002	0.009	<0.001
23. Contains no artificial ingredients	3.71 (1.106)	3.75 (1.092)	3.78 (1.066)	3.84 (1.038)	0.011	3.52 (1.180)	3.52 (1.185)	1.000	<0.001	<0.001
**Factor 6—Price**										
6. Is not expensive	3.38 (1.090)	3.48 (1.107)	3.46 (1.093)	3.56 (1.107)	<0.001	3.17 (1.054)	3.25 (1.079)	0.021	<0.001	<0.001
36. Is cheap	3.02 (1.125)	3.19 (1.188)	3.08 (1.132)	3.26 (1.189)	<0.001	2.84 (1.086)	3.02 (1.171)	<0.001	0.001	0.002
12. Is good value for money	4.01 (1.017)	4.00 (0.998)	4.06 (0.990)	4.06 (0.961)	0.896	3.87 (1.079)	3.81 (1.069)	0.152	0.004	<0.001
**Factor 7—Weight Control**										
3. Is low in calories	3.06 (1.069)	3.29 (1.067)	3.07 (1.064)	3.32 (1.040)	<0.001	3.03 (1.087)	3.22 (1.136)	<0.001	0.579	0.139
17. Helps me control my weight	3.46 (1.167)	3.57 (1.113)	3.50 (1.145)	3.61 (1.087)	<0.001	3.37 (1.220)	3.48 (1.180)	0.012	0.105	0.087
7. Is low in fat	3.17 (1.045)	3.31 (1.072)	3.19 (1.046)	3.33 (1.067)	<0.001	3.14 (1.046)	3.25 (1.087)	0.012	0.541	0.228
**Factor 8—Familiarity**										
33. Is what I usually eat	3.43 (1.080)	3.45 (1.094)	3.45 (1.088)	3.48 (1.118)	0.278	3.37 (1.058)	3.35 (1.025)	0.730	0.219	0.061
8. Is familiar	3.48 (1.093)	3.53 (1.040)	3.49 (1.087)	3.57 (1.043)	0.007	3.47 (1.115)	3.42 (1.027)	0.346	0.745	0.026
21. Is like the food I ate when I was a child	2.87 (1.197)	3.00 (1.180)	2.91 (1.196)	3.04 (1.187)	<0.001	2.75 (1.191)	2.88 (1.152)	0.003	0.044	0.031
**Factor 9—Ethical Concern**										
20. Comes from countries I approve of politically	2.18 (1.176)	2.39 (1.227)	2.23 (1.183)	2.44 (1.240)	<0.001	2.04 (1.148)	2.26 (1.184)	<0.001	0.009	0.023
32. Has the country of origin clearly marked	3.29 (1.299)	3.30 (1.312)	3.37 (1.299)	3.38 (1.318)	0.878	3.06 (1.274)	3.08 (1.267)	0.641	<0.001	<0.001
19. Is packaged in an environmentally friendly way	3.18 (1.154)	3.24 (1.180)	3.29 (1.118)	3.33 (1.157)	0.140	2.87 (1.191)	3.00 (1.206)	0.009	<0.001	<0.001

*n*, number of participants; SD, standard deviation; ^a^ the numbers in front of the items represent the ordinal numbers of the items in both the original version of the Food Choice Questionnaire developed by Steptoe et al. [24] and the Western Balkan Countries version of the Food Choice Questionnaire developed by Milošević et al. [23]; ^b^ within-group differences before and during the COVID-19 pandemic were calculated using ANOVA with repeated measures; ^c^ between-group differences for both before and during the COVID-19 pandemic were calculated using a one-way ANOVA. The items were included within nine factors as indicated. The items and factors are presented in order that corresponds to the original Food Choice Questionnaire [24].

**Table 4 nutrients-13-03165-t004:** The factors influencing food choice before and during the COVID-19 pandemic for the total sample and separately with respect to sex.

Factors	Total Sample (*n* = 1232)		Women (*n* = 890)			Men (*n* = 339)			Women vs. Men before COVID-19 Pandemic	Women vs. Men during COVID-19 Pandemic
	Before COVID-19 Pandemic Mean (SD)	During COVID-19 Pandemic Mean (SD)	Before COVID-19 Pandemic Mean (SD)	During COVID-19 Pandemic Mean (SD)	*p* ^a^	Before COVID-19 Pandemic Mean (SD)	During COVID-19 Pandemic Mean (SD)	*p* ^a^	*p* ^b^	*p* ^b^
Health	3.84 (0.856)	3.91 (0.862)	3.90 (0.836)	3.99 (0.833)	<0.001	3.68 (0.887)	3.71 (0.903)	0.193	<0.001	<0.001
Mood	3.37 (0.928)	3.36 (1.013)	3.41 (0.919)	3.41 (1.006)	0.742	3.25 (0.942)	3.21 (1.018)	0.133	0.009	0.002
Convenience	3.84 (0.858)	3.90 (0.888)	3.93 (0.833)	4.01 (0.858)	<0.001	3.60 (0.878)	3.61 (0.902)	0.657	<0.001	<0.001
Sensory Appeal	3.85 (0.849)	3.85 (0.877)	3.92 (0.840)	3.93 (0.860)	0.479	3.67 (0.847)	3.62 (0.884)	0.091	<0.001	<0.001
Natural Content	3.77 (0.929)	3.81 (0.947)	3.83 (0.908)	3.89 (0.904)	0.002	3.61 (0.954)	3.60 (1.014)	0.733	<0.001	<0.001
Price	3.47 (0.905)	3.56 (0.942)	3.53 (0.899)	3.63 (0.930)	<0.001	3.30 (0.900)	3.36 (0.947)	0.009	<0.001	<0.001
Weight Control	3.23 (0.917)	3.39 (0.954)	3.25 (0.911)	3.42 (0.942)	<0.001	3.18 (0.937)	3.32 (0.987)	<0.001	0.256	0.095
Familiarity	3.26 (0.883)	3.32 (0.921)	3.28 (0.884)	3.36 (0.925)	<0.001	3.20 (0.880)	3.22 (0.902)	<0.001	0.123	0.012
Ethical Concern	2.88 (0.992)	2.97 (1.040)	2.97 (0.991)	3.05 (1.038)	<0.001	2.66 (0.961)	2.78 (1.019)	<0.001	<0.001	<0.001

*n*, number of participants; SD, standard deviation; ^a^ within-group differences before and during the COVID-19 pandemic were calculated using ANOVA with repeated measures; ^b^ between-group differences for both before and during the COVID-19 pandemic were calculated using a one-way ANOVA. The factors are presented in order that corresponds to the original Food Choice Questionnaire [24]. The average mean score was calculated for all the factors.

## Data Availability

The data presented in the study are available on request from the corresponding author. The data are not publicly available due to privacy issues.

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
