# Peer review of "Evaluation of the Food Choice Motives before and during the COVID-19 Pandemic: A Cross-Sectional Study of 1232 Adults from Croatia"

_nutrients, 2021, doi:10.3390/nu13093165_

Round 1

Reviewer 1 Report

This paper investigates changes of food choice motivation before and during the pandemic. The authors use a Croatian sample of adults. The paper deals with an important and very timely topic and the paper will be of great interest to the jounral's readers. Overall, it is a good and well-written paper. I list a few minor comments below.

Title: "changes in food choice" may be misleading as the authors measured food choice motivation and not actual food choice.

Introduction:

“The inconsistency of the results on the impact 56 of the COVID-19 pandemic on the diet quality was also confirmed by a scoping review 57 conducted by Bennett et al. [9]. The limitation among these studies lies in the fact that 58 changes in diet quality are just a consequence of an actual change in the food choice motivations, and this latter is still not elucidated.”

It is not entirely clear what the authors mean by “food choice motivations”. An important point missing here is the fact that individuals have been affected differently by the pandemic. While some were able to work remotely, others were not. While for some the workload has increased massively, for others it has not. Authors should add this information to their argumentation.

Either in the introduction or in the method section, authors should describe what the Food Choice Questionnaire is, what it measures and the rationale for using it here in this study.

Method:

I understand that this study was part of an international study conducted in Croatia and Belgium. The authors should explicitly state that only data from Croatia is considered here. This piece of information seems to be missing.

What do the authors mean by “potential benefits”?

Was there any incentive for study participation? Did participants receive payment?

Please specify what you mean by “information on COVID-19”. It is currently unclear what kind of questions participants had to answer.

Please elaborate what the factor analysis was used for. When reading the methods section, it is not entirely clear why this analysis was chosen. Furthermore, given that the authors used an existing and validated questionnaire, I really wonder why they did not just use or apply the existing factors but instead run an exploratory factor analysis.

Discussion:

“Since the very beginning of the COVID-19 pandemic, the importance of proper nutrition, as a natural booster of the immune system, has been widely emphasized [32-35].”

I agree with the authors that a healthy nutrition is an important basis for the immune system. However, given the current trend to picture nutrition as medication, I strongly recommend rephrasing. In my view, proper nutrition is a requirement for a healthy immune system (and not a booster).

Line 394-399: This paragraph can be deleted. Discussing the rationale of the study and why only data from Croatia is presented here does not belong here. Authors can elaborate in the method section but this should not be part of the discussion.

Line 432-438: While I support the basic idea here, the authors are very general. Please be more specific. For instance, it remains unclear what the authors mean by “disinformation”. What is mainly missing here is some specific reference to your study or your findings.

Recall bias: needs to be explained

Reviewer 2 Report

The article describes a study conducted on a large sample of Croatian adults to evaluate the motives of food choices before and during the COVID-19 pandemic. Considering the importance of a correct diet to cope more adequately with the COVID-19 Disease, the study appears to be of great interest not only for scientists but also for policy makers and other stakeholders in the field of health and food education. The identified changes in dietary habits can provide useful information both to understand individual behaviors and to guide educational actions and health policies in the field of prevention.

The article is very clearly written and is understandable and pleasant to read. According to the authors themselves, this is an original study as there are few other studies that focus on food motivations during the pandemic.

I have a general suggestion and some minor revisions requested.

As a general suggestion, I would ask the authors to spend some time explaining the specific situation of Croatia. For example, in the methodology section, was the data collection period carried at a time of lockdown or after? In the discussion section, I would comment the results of the study on the base of previous studies on the Croatian population. For example, the authors cite the need to replicate the study internationally, to take into account socioeconomic and cultural diversity: in this sense, it would be interesting to understand the Croatian situation. Is the whole population obese or overweight, is the country a rich nation, what about its food culture? Which actions of diet-related disease prevention are implemented? And what food education actions?

Line 28: are the quotes (“….”) really necessary? The same at lines 438, 471-472 and 489-490. Otherwise, please add quote.

Line 29: due to the complexity of interpretations of the term “sustainability”, I would avoid using it here. It is not necessary. Same suggestion at lines 436 and 469 where I suggest to use something like “long-lasting” or similar.

Line 36: I suggest inserting a concise definition of non-communicable disease. You can also move here what is written in brackets at lines 428-429.

Line 45: you start talking about Croatia and then, at lines 47-48, you go back to talking about a general situation. Since what has been said at line 45 has occurred globally, I would change the sentence to: various types and extents of COVID-19 lockdown have occurred all over the world, resulting in…” or similar.

Line 60: this concept seems to be crucial for purpose of the article, but it needs to be expanded and better explained.

Line 86: do not repeat “socio” both in sociocultural and socioeconomic. Use just cultural. Same thing at line 406.

Lines 101-106: I suggest moving this paragraph to the Acknowledgments section (Line 510) where I would cancel “The authors would like to thank al the study participants”. The participants have given their informed consent but not their endorsement.

Line 109: please, add if at that time Croatia was in lockdown or not.

Line 113: please explain which relevant groups and social media networks.

Lines 109-115: why did you choose to have a sample strongly represented by people close to the socio-economic and cultural context of the research team members, who invited the participants through their professional networks? Are there other studies that have made the same methodological choice? Please, cite them or explain your choice to have a non-probabilistic sample.

Line 126: I prefer Ethical issues rather than Ethics.

Lines 130-131: please, explain which potential benefits.

Lines 138-139: is this sentence really necessary?

Lines 140-141: I suggest moving the first part of this paragraph (until GDPR 2016/679) to the Informed Consent Statement, at line 507. Then, please delete the rest of the paragraph that repeats what is written at lines 503-505.

Line 153: define the acronym BMI.

Lines 218-221: this is repeated in the note to table 1, lines 238-239, please, eliminate the duplication.

Line 269: please, leave a space after the table.

Lines 376-377: this can be explained by other reasons, for example it may be that men have worked more than women outside the home, even during the lockdown, and have come into contact with more people with a greater risk of getting infected. I would remove this sentence or support it with literature.

Lines 394-399: I would move this paragraph to section 4.2., line 466, saying that the intention is to continue comparing the results with those obtained in Belgium.

Line 406: cultural.

Line 406: medical,

Lines 406-412: this paragraph is not clear, please explain it better.

Lines 413-416: I'm not sure if this can be said. Based on your results it seems that the environmental or ethical motivations are not that decisive. Here too, it should be better explained and supported with literature.

Line 438: please, give here some information about what is being done in Croatia in the field of food education and / or prevention of diet-related diseases or information and examples of such successful educational programmes and strategies.

Lines 466-467: please, cancel this sentence and insert and expand the paragraph at lines 394-399.

Line 502: it seems to contradict the fact that the study was conducted within the CFC CRO-BE project.
